# Time-Dependent Frictional Properties Of Granular Materials Used In Analogue Modelling: Implications For Mimicking Fault Healing During Reactivation And Inversion

Michael Rudolf[1,2], Matthias Rosenau[2], and Onno Oncken[2]

[1]Institute of Applied Geosciences, Fachgebiet Ingenieurgeologie, Technical University Darmstadt, Schnittspahnstraße 9, 64287 Darmstadt, Germany
[2]Lithosphere Dynamics, Helmholtz Centre Potsdam, GFZ German Research Centre for Geosciences, Telegrafenberg, 14473 Potsdam, Germany

**Correspondence:** Michael Rudolf (rudolf@geo.tu-darmstadt.de)

**Abstract.** Analogue models are often used to model long-term geological processes such as mountain building or basin inversion. Most of these models use granular materials such as sand or glass beads to simulate the brittle behaviour of the crust. In granular material, deformation is localised in shear bands, which act as an analogue to natural fault zones and detachments. Shear bands, also known as faults, are permanent anomalies in the granular package and are often reactivated during a test run. This is due to their lower strength compared to the undeformed material. When the fault movement stops, time-dependent healing immediately begins to increase the strength of the fault. Faults that have been inactive for a long time therefore have a higher strength than younger faults. This time-dependent healing, also called time consolidation, can therefore affect the structure of an analogue model as the strength of the fault changes over time. Time consolidation is a well-known mechanism in granular mechanics, but it is poorly described for analogue materials and on the time scales of typical analogue models. In this study, we estimate the healing rate of different analogue materials and evaluate the impact on the reactivation potential of analogue faults. We find that healing rates are generally less than 3 % per tenfold increase in holding time, which is comparable to natural fault zones. We qualitatively compare the frictional properties of the materials with grain characteristics and find a weak correlation of healing rates with sphericity and friction with an average quality score. Accordingly, in models where there are predefined faults or reactivation is forced by blocks, the stability range of the reactivatable fault angles can decrease by up to 7° over the duration of 12 hours. The stress required to reactivate an existing fault can double in the same time, which can favour the development of new faults. In a basin inversion scenario, normal faults cannot be inverted because of the strong misorientation, so time consolidation plays little additional role for such models.

## 1 Introduction

Crustal tectonics involves fault localization and reactivation as well as changes in fault slip rates (transients) and slip directions (inversion) over timescales from hundreds of years (seismic cycle) to tens of millions of years (Wilson cycle). For example, the inversion of sedimentary basins is governed by a change from extensional to compressional tectonics, which leads to the shortening of basin structures (Turner and Williams, 2004). In such a scenario, shortening can be accommodated by either

the reactivation of pre-existing structures or the formation of new faults. Structural inhomogeneities that localise deformation during inversion are mainly the normal faults formed under extension and stratigraphic layers of the basin sediments. In many cases, newly formed structures link with existing ones forming shortcuts to create the energetically most favourable fault configuration. This interplay of extensional and compressional tectonics as well as inherited and newly formed structures creates a complex structural inventory that is inherent to many sedimentary basins world-wide, which are regions that are of high societal and economic importance hosting earthquakes, ore deposits and hydrocarbon resources (Buchanan et al., 1995; Turner and Williams, 2004).

Many experimental studies show that the localization of brittle deformation into narrow bands is governed by the strain weakening characteristics of rocks. As a result, faults are mechanically weaker than the undeformed host rock and thus are reactivated when subjected to stress (Sibson, 1980). However, other factors influence the reactivation characteristics of a fault, most notably the orientation of stresses with respect to the fault. Additionally, fluid pressure, the exact mechanical properties of the fault zone and interaction with other faults can prevent or promote fault reactivation (Niemeijer et al., 2008). There is a multitude of numerical, mechanical and analogue modelling studies that address the influence of these parameters on fault reactivation (e.g. Jara et al., 2018; Yagupsky et al., 2008; Panien et al., 2006, and references in Table 2). Especially in analogue models, the reactivation of normal faults is largely dependent on the orientation as the other factors are rarely incorporated into the model (Bonini et al., 2012).

Analog modelling is a widely used technique for tectonic modelling in general and for basin inversion models in particular because it is inherently three-dimensional and can handle discontinuities with large displacements, which is a challenge for most numerical approaches. Analogue models are built around the principle of similitude (Hubbert, 1937), which states that a system can be modelled by a geometrically smaller model if the governing dimensionless properties are the same. This is frequently used for geological modelling of complex tectonic processes and in other disciplines where numerical approaches are still not entirely feasible, such as hydraulic engineering. While numerical models are easy to quantify and there is a large freedom in defining material properties, analogue modellers have access to only a small range of suitable materials for specific problems. In the past this has been limited to various natural (e.g. sands) and artificial granular materials (e.g. glass beads) as frictional components (Klinkmüller et al., 2016; Ritter et al., 2016a, 2018; Montanari et al., 2017), sometimes mixed with more fine-grained (powder) materials, like flour and plaster, to increase the cohesion according to scaling laws (Poppe et al., 2021; Galland et al., 2006). Models involving viscous layers (e.g. lower crust, salt) typically use silicone oils (PDMS) (Rudolf et al., 2016; Weijermars, 1986) and other visco-elastic materials. Recently, there is a surge in new materials to fine-tune specific properties of the brittle or ductile layer (see Reber et al., 2020, for a detailed review). However, all analogue models rely on accurate and suitable material characterisation to be able to quantify the similarity of stresses and ultimately the similitude of the model (ten Grotenhuis et al., 2002).

A property that has received little to no attention is the time dependency of frictional material properties, in particular healing (i.e. static strengthening). In this context we identify two major implications for the analysis of reactivated structures with analogue models. The first is that due to the consolidation of granular materials over time, the reactivation strength increases. This could lead to problems with repeatability when there are differences in the timing of extensional and compressional

phases between model runs. In the engineering community this effect is known as time consolidation and usually is considered as being minor in typical analogue materials (Schulze, 2008). However, most studies on time consolidation focus on large piles or silos with several meters overburden material and not a few centimetres as in analogue models. Due to the increase of shear resistance over time, reactivation angles and structures could differ between a model that has for example a one minute static phase between extension and compression in comparison to a model that has a several hours stop between extension and compression. Models with erosion and sedimentation are especially prone to this effect because they usually require the model to be stopped for some time to add or remove material. The same applies to models that have to be measured using time-consuming procedures, such as laser scanners or CT scanners. Secondly, natural faults also show time dependent healing due to pressure solution, Ostwald ripening and fracture sealing by hydrothermal minerals (Karner et al., 1997; Niemeijer et al., 2008). At short time scales of the seismic cycle, this behaviour is described by the dimensionless healing rate $b$ in the rate-and-state framework (Dieterich, 2007) while for longer tectonic time scales the quantification is more challenging (Yasuhara et al., 2005). To accurately mimic this natural healing in analogue models it is required that the analogue materials show quantitatively similar characteristics as rocks, e.g. the same healing rate $b$.

Consequently, the aim of this study is: (1) to quantify the healing properties of analogue materials from different laboratories and to relate them to first order observable grain characteristics, (2) to analyze the impact on the reactivation angles during typical analogue models of basin inversion, and (3) to examine the possible use of certain materials as analogues for naturally healing faults. The material properties of 36 samples were characterised by standardised ring-shear tests and slide-hold-slide tests. Furthermore, we used image analysis to gather more detailed information on grain size distributions, grain shapes and grain surface features for each material. The results are then used to review typical analogue modelling schemes and to probe possible scenarios under which the materials may or may not be suitable for modelling.

## 2 Methodology

### 2.1 Materials

For this study we use a selection of samples that were archived from collaborative material properties studies (e.g. Klinkmüller et al., 2016)in our laboratory and newly acquired samples from 14 laboratories worldwide that sent materials that are currently used for their analogue models. Some of these materials were characterised in terms of time-independent frictional properties in the original GeoMod benchmark study by Klinkmüller et al. (2016) while others have been characterised in the framework of EPOS transnational access activities (Table 1, Wessels et al. (2022); Elger et al. (2022)). Most of them have been subjected to standard ring shear tests to determine their time-independent friction parameters (friction coefficient and cohesion) and rate-dependent (velocity strengthening/weakening) friction parameters. To have consistent evaluation parameters, we re-picked and re-analysed the existing datasets with the newest software RST-Evaluation (Rudolf and Warsitzka, 2021), which is the current standard for ring shear tests in our laboratory. New materials that were not in our test database were tested with the standard ring-shear test outlined in section 2.2. For this study, we tested all samples with the slide-hold-slide procedure developed in Rudolf et al. (2021),which is outlined in section 2.4.

**Table 1.** Samples included in this study and references where standard ring shear tester data can be found. Materials that were included in Klinkmüller et al. (2016) are highlighted with '∗'. Some materials have not been published previously and are marked with 'new'. These have been tested for friction and healing properties. All other samples have only been tested for their healing properties. Sample numbers refer to the numbering scheme of the slide-hold-slide tests found in the data publication.

| No. | | Lab of origin and description | Ref. for previous ring shear test data and results |
|---|---|---|---|
| **Quartz Sands** | | | |
| 02 | | Bern, 2015 | Schmid et al. (2020b); Zwaan et al. (2018) |
| 08 | ∗ | Bern, 2008 | Klinkmüller et al. (2016) |
| 28 | | Bern CarloAG, 100-300 $\mu m$ | Schmid et al. (2020b); Zwaan et al. (2018) |
| 03 | | Utrecht, 2018 | Willingshofer et al. (2018) |
| 05 | | Rennes NE34, 2021 | new (B. Guillaume) |
| 06 | ∗ | Parma | Klinkmüller et al. (2016) |
| 07 | ∗ | Kyoto | Klinkmüller et al. (2016) |
| 10 | ∗ | Uppsala | Klinkmüller et al. (2016) |
| 11 | ∗ | Cergy Pontoise | Klinkmüller et al. (2016) |
| 14 | ∗ | RHU-London | Klinkmüller et al. (2016) |
| 15 | ∗ | GFZ G12, $<400\,\mu m$ | Klinkmüller et al. (2016); Rosenau et al. (2018a); Ritter et al. (2016b) |
| 16 | ∗ | GFZ G23, $<630\,\mu m$ | Klinkmüller et al. (2016); Rosenau et al. (2018b); Ritter et al. (2016b) |
| 17 | | Prague, ST55 | Warsitzka et al. (2019b) |
| 40 | ∗ | Prague | Klinkmüller et al. (2016) |
| 36 | | Lille, G39 | new (B. Vendeville) |
| 37 | ∗ | Lille, NE34 | new (B. Vendeville) and Klinkmüller et al. (2016) |
| 41 | ∗ | Wroclaw | Klinkmüller et al. (2016) |
| 42 | ∗ | Taiwan | Klinkmüller et al. (2016) |
| 43 | ∗ | Ouro Preto | Klinkmüller et al. (2016) |
| 46 | | Freiburg | Aschauer and Kenkmann (2017) |
| **Corundum Sands** | | | |
| 12 | | Bern, 2019 | Schmid et al. (2020a) |
| 24 | ∗ | Bern, white | Klinkmüller et al. (2016) |
| 45 | ∗ | Bern, brown | Klinkmüller et al. (2016) |
| 09 | | GFZ NKF120 | new |
| **Feldspar Sands** | | | |
| 13 | | Utrecht | Willingshofer et al. (2018) |
| 35 | | Bern FS900S | Zwaan et al. (2022) |
| **Garnet Sands** | | | |
| 18 | ∗ | GFZ, 2008 | Klinkmüller et al. (2016) |
| **Glass Beads** | | | |
| 21 | | GFZ, 40-70 $\mu m$ | Pohlenz et al. (2020a) |
| 19 | | GFZ, 70-110 $\mu m$ | Pohlenz et al. (2020b) |
| 22 | ∗ | GFZ, 100-200 $\mu m$ | Klinkmüller et al. (2016) |
| 20 | ∗ | GFZ, 300-400 $\mu m$ | Klinkmüller et al. (2016); Pohlenz et al. (2020c) |
| 25 | | Prague, 0-50 $\mu m$ poured | (Rudolf et al., 2022) |
| 26 | | Prague, 100-200 $\mu m$ | Rosenau et al. (2022) |
| 38 | | Lille CVP, 70-110 $\mu m$ | new (B. Vendeville) |
| 39 | | Lille Eyraud, 70-110 $\mu m$ | new (B. Vendeville) |
| **Zircon Sands** | | | |
| 23 | ∗ | GFZ, 2009 | Klinkmüller et al. (2016) |
| **Foam Glass** | | | |
| 47 | | GFZ, 250-500 $\mu m$ | Warsitzka et al. (2019a) |

## 2.2 Ring-shear Tester Setup

We use a RST-01.pc (Schulze, 1994) ring-shear tester to measure the frictional properties of the dry granular materials. The method is well established for analog materials (Lohrmann et al., 2003; Panien et al., 2006; Klinkmüller et al., 2016; Montanari et al., 2017) and follows international standards for powder and bulk material testing (ASTM, 2016). The machine consists of a rotating, ring-shaped shear cell onto which normal stress is applied using a stationary lid. The shear stress required to hold the lid in place is measured using two tie rods that are each attached in series to force transducers. To improve the contact of the cell and lid with the material, the surface of the lid, as well as the bottom of the shear cell is structured with slats and grooves. Normal stress is applied through a cantilever system with a moving mass and therefore is instantly adjusted by gravity, in contrast to other mechanical tests where the stresses are adjusted with a servo-hydraulic or electronic system. During each measurement, the shear velocity, shear stress, normal stress and lid position is monitored. Further information on the setup and device is available in Schulze (1994), Lohrmann et al. (2003) and Ritter et al. (2016a).

The testing procedure follows standardised procedures for sample preparation (Lohrmann et al., 2003; Klinkmüller et al., 2016), testing procedure (Lohrmann et al., 2003; Ritter et al., 2016a) and data analysis (Rosenau et al., 2018a; Rudolf et al., 2021). All samples are first oven dried to remove excess humidity and then stored in the air-conditioned laboratory for several days to equilibrate with ambient laboratory conditions of $T = 25\,°C$ and $\approx 50\,\%$ humidity. The samples are sieved using the SM sieve into the cell from a height of ca. 30 cm and above, ensuring a similar package density for each test (Lohrmann et al., 2003). We do not use the same sieve as Klinkmüller et al. (2016), called 'GeoMod'-sieve because for our measurements the samples are pre-sheared and the package density after sieving is not a primary concern. Excess material is scraped off and the weight of the material is determined. After inserting the cell into the tester, the normal stress is applied and the shear procedure is started. Each property has a specific shear procedure, which is outlined in the respective subsections (Section 2.3 for friction $\mu$ and cohesion $C$, Section 2.4 for the healing rate $b$).

## 2.3 Mohr-Coulomb Friction

Most tectonic analogue models use dry granular materials as analogues for crustal rocks in the brittle regime (Klinkmüller et al., 2016) with only a few exceptions that use wet clay or other non granular material (Bonini et al., 2012). The greatest advantage of granular materials is that they obey the empirical Mohr-Coulomb criterion (Equation 1):

$$\tau = \mu\sigma + C \tag{1}$$

Granular analogue materials show friction coefficients $\mu$ and cohesions $C$ that are comparable (in case of $C$ when scaled) to typical crustal rocks: In particular sands with $\mu = 0.6$ to $0.7$ and $C$ in the order of tens of Pa (scaling to few MPa) or glass beads with $\mu = 0.4$ to $0.5$ and $C$ in the order of few Pa. Moreover, the granular materials show stress-strain relationships similar to crustal rocks involving strain weakening and static healing (Lohrmann et al., 2003; Ritter et al., 2016a). This gives rise to three different coefficients of friction $\mu$ and cohesion $C$ attributed to different stages of fault evolution (Lohrmann et al., 2003):

The highest strength $\mu_{peak}$ is reached during initial shearing of undisturbed granular material and therefore is analogous to the strength of undeformed rock (static friction). With continued shearing the materials looses strength (strain weakening) and reaches a lower strength during stable sliding $\mu_{stable}$ corresponding to the sliding resistance of a fault zone (sliding, dynamic or kinetic friction). If a material has been sheared the granular fault zone is persistent and leaves a heterogeneity with lower density in the bulk material. If re-sheared, a new peak strength $\mu_{reactivation}$ occurs that reflects the strength of a pre-existing fault zone. Usually, it is higher than the stable sliding strength but lower than the initial peak strength. Consequently, models with granular materials localise deformation into narrow shear bands because of their lower sliding resistance and reactivate these structures under favourable circumstances due to the lower reactivation strength. From now on we will refer to these three friction coefficients by their shortened versions $\mu_p$, $\mu_s$ and $\mu_r$ throughout this study. Note that corresponding cohesions $C_p$, $C_s$ and $C_r$ exist.

## 2.4 Time Consolidation (healing) and Rate-and-State Friction

In addition to the dependence on effective normal stress $\sigma$, sliding and reactivation friction coefficients $\mu_s$ and $\mu_r$ show a measurable dependence on slip rate $\dot{\delta}$ and hold time $t_h$, respectively. This is highly non-linear and described using the rate-and-state framework (Dieterich, 1978). This formulation is widely accepted as a good heuristic approximation of laboratory shear tests and natural phenomena (Marone, 1998; Scholz, 2002; Dieterich, 2007) where a time and strain rate dependence is observed. In general, there are two additional contributions to shear resistance: the rate effect $a\,ln\frac{\dot{\delta}}{\delta^*}$ and the state effect $b\,ln\frac{\theta}{\theta^*}$. Both are defined by a ratio with respect to reference constants (denoted by asterisks) and added to the reference friction $\mu_0$:

$$\tau = \sigma\left[\mu_0 + a\,ln\frac{\dot{\delta}}{\delta^*} + b\,ln\frac{\theta}{\theta^*}\right] \tag{2}$$

The direct effect $a$, healing rate $b$ and $\mu_0$ are derived empirically from experimental measurements. The evolution of state $\theta$ can take several forms as a function of time (aging law, Dieterich, 1978), slip (slip law, Ruina, 1983), time dependent healing (Kato law, Kato and Tullis, 2001) or stressing rate (Nagata law, Nagata et al., 2012), which has to be assessed from experiments.

The change of strength over time in granular materials is known as time consolidation and can be very large depending on the material (Schulze, 2008). For this study we assume purely time dependent healing (ageing law) and use the healing rate $b$ to calculate the time-consolidation for each sample. We use slide-hold-slide tests to measure the healing rate following the procedure outlined in Rudolf et al. (2021, and references therein). After sample preparation (Section 2.2) and loading with $\sigma_N = 1\,kPa$ the sample is sheared by 10 mm at a loading velocity $v_L = 0.5\,\frac{mm}{s}$ leading to a fully developed shear zone. Then the sample is subjected to several slide-hold-slide intervals (Figure 1a). The hold intervals are increased exponentially from $t_h = 10^1$ to $10^4\,s$ at increments of half a tenfold increase in time and repeated three times per interval. An additional, single hold interval of $t_h = 36000\,s = 10\,h \approx 10^{4.55}\,s$ gives a long term data point to each series. The full test duration is approximately 23 hours.

We pick the shear stress needed to reactivate the shear zone after each hold time $t_h$ and normalise it to the mean stress during stable sliding: $\Delta\mu = \mu_r - \bar{\mu}_s$ (Figure 1b). This results in an effective stress measure assuming no cohesion. The healing rate is

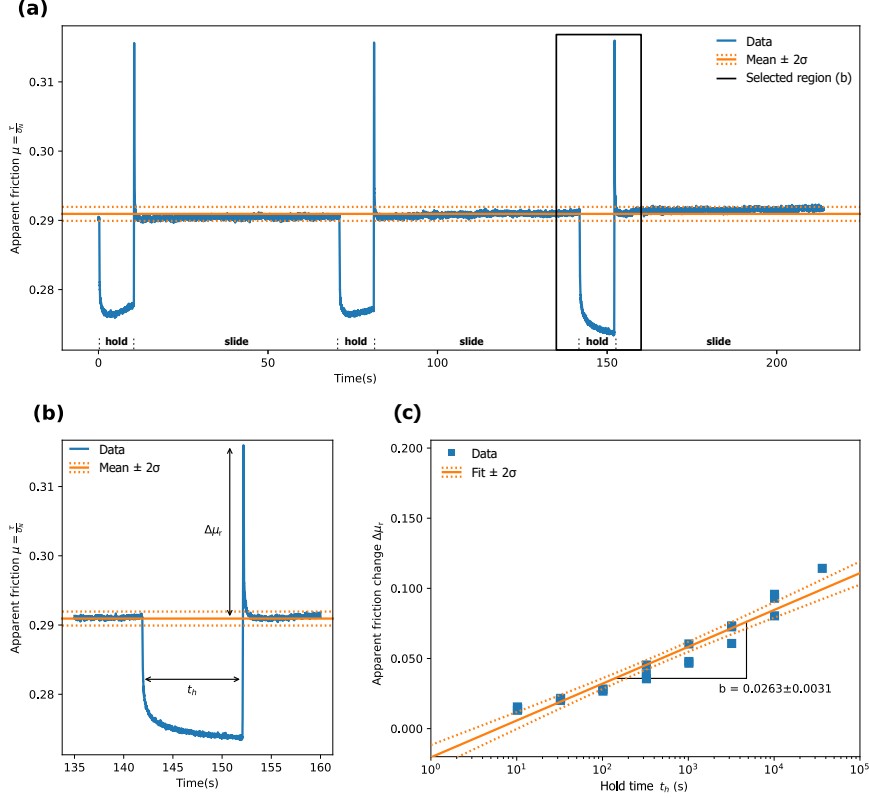

**Figure 1.** Time-series of one slide-hold-slide sequence and picked values. (a) Slide-hold-slide sequence of GFZ 70-110 $\mu m$ glass beads for $t_h = 10\,s$. Slide and hold phases are marked along the time axis. (b) Detailed time-series showing a single hold phase and the picked value for $\Delta\mu_r$. (c) Fit of all picked reloading friction values versus hold time. The error ranges are the 95 % confidence bands for the fit.

then the change of $\Delta\mu$ in comparison to the natural logarithm of hold time $t_h$ (Beeler et al., 1994; Bhattacharya et al., 2017) and is obtained from as the slope of $\Delta\mu$ vs. $ln\,t_h$ (Figure 1c):

$$b = \frac{\delta\Delta\mu}{\delta\,ln\,t_h} \tag{3}$$

We calculate the compaction rate in the same manner by using the difference in lid position between the start and end of the hold phase. The same power-law relation is found with strong compaction for short hold phases. Values for compaction rate are negative because higher compaction results in smaller sample heights. All data is automatically picked and evaluated using a dedicated Python code published open source in the software "RST-Stick-Slipy" (Rudolf, 2021).

**Table 2.** Categorization scheme for granular materials with a simple 'quality score'. A lower score indicates a tendency for higher internal friction and time consolidation.

| Parameter | Definition | Weight | Explanation | References |
|---|---|---|---|---|
| Sphericity | 4 → ● = perfect <br> 3 → ● = high <br> 2 → ◖ = medium <br> 1 → ▬ = low | 0.3 | Aspherical particles lead to higher friction. | Härtl and Ooi (2011); Chen et al. (2022) |
| Roundness | 4 → ● = rounded <br> 3 → ◕ = subrounded <br> 2 → ◆ = subangular <br> 1 → ◆ = angular | 0.9 | Round particles tend to create lower bulk friction. Angular particles favour stable sliding, while spherical particles exhibit stick-slip. | Mair et al. (2002); Suh et al. (2017) |
| Surface roughness | 4 → ‾ = smooth <br> 3 → ⌣ = slightly rough <br> 2 → ∿ = shelly <br> 1 → ⌄ = rough/jagged | 1.8 | Higher roughness leads to higher inter-particle friction resulting in higher bulk friction. Higher dilation needed for rough materials. | da Cruz et al. (2005); Tapia et al. (2019) |

## 2.5 Grain Characteristics

Ultimately the frictional response of a bulk material is the result of granular interactions and therefore depends primarily on the geometric characteristics of the grains. The shear zones forming in the ring-shear tester usually span 11 to 16 times the mean grain size (Panien et al., 2006). The grains react to stress by creating force chains that frequently change their orientation (Cates et al., 1998; Daniels and Hayman, 2008). As a result, frictional resistance between individual grains and material elasticity has major implications on the bulk behaviour.

However, it is technically very challenging to measure friction between the individual grains, and therefore we assess the tendency of each material to create locked states by using a qualitative index for several key features. We categorise the materials with three different parameters: sphericity, roundness and surface roughness. The definition of the shape parameters follows that of clastic sediments. Sphericity is defined as the closeness to a perfect sphere. Roundness is given by the shape of the edges of a grain. Surface roughness is a measure of the small scale surface properties, not of the grain shape itself. Each parameter is assigned a score from 1 to 4 that expresses the materials proneness to locking with 4='low impact' and 1='high impact'. Some materials have grains with different characteristics, e.g., sands that are crushed are usually mixtures of rounded and angular grains. Therefore, depending on the degree of mixing, they are given an intermediate quality score. Grain size distribution was not taken into account because comparable measurements do not exist for all materials. A heterogeneous grain size distribution changes the bulk density of the material, which can influence the frictional characteristics (Lohrmann et al., 2003). However, the effect is minor (Mair et al., 2002). Table 2 shows an overview of the parameters, criteria and associated literature.

The quality score is a qualitative measure that only gives a general tendency of material behaviour and follows an arbitrary scale. We use this scale to turn a qualitative observation into a quantity. Here, the purely subjective observation of the grains, without capturing a truly measurable quantity, represents a major source of error. The quality score is therefore only a theoretical support to find possible correlations between the parameters. In no way should the score be seen as an objective quantity. It could serve as a preliminary stage for future measurable variables and give a rough direction.

However, the relative importance of the grain parameters sphericity, roundness and surface roughness can be estimated by comparing studies that systematically varied them as part of either numerical or experimental studies (references in Table 2). To account for the impact of each parameter we take a weight average that uses the estimated effect on bulk friction from the references. We assume that this weighted average tends to reflect the amount of inter-particle locking during a hold phase and therefore also is a proxy to healing rate $b$. The weights are estimated by comparing the influence of each parameter on the value of friction in the respective studies. Härtl and Ooi (2011) report an increase of sliding friction from $\mu_d = 0.56$ for spherical particles to $\mu_d = 0.59$, which is a 10 % increase in friction. Particle roundness has been shown by Mair et al. (2002) to increase friction from $\mu = 0.45$ for round particles to $\mu_d = 0.6$ that is an increase of 30 %. Tapia et al. (2019) show an increase from $\mu_d = 0.23$ for slightly roughened to $\mu_d = 0.37$ for highly roughened particles reflecting an 60 % increase. This is the strongest influence of all examined parameters. As a result, the relative weighting of the parameters is given as the ratio between the three percentages to each other summed up to three. Coincidentally, all three percentages sum up to one and therefore the percentages given previously, also reflect their relative proportion for the weighted average (10 %, 30 % and 60 % or 0.3, 0.9, 1.8 given as weights).

## 2.6   Reactivation of Faults

The reactivation of pre-existing faults in nature as well as in analogue models is primarily governed by (a) fault geometry, (b) the surrounding stress field and the (c) fault's frictional properties (Bonini et al., 2012). For natural fault systems an additional mechanism is fluid overpressure e.g. in basin sediments. However, fluids and fluid pressures cannot be accurately modelled in a scaled fashion and is thus rarely implemented in tectonic analogue models. We therefore focus on the aforementioned three factors (a) to (c) to estimate the tendency to reactivate a pre-existing fault instead of forming a new fault. We do this in a typical basin inversion scenario and use a simplified Amonton wedge model to estimate the stresses needed to push a sidewall with a reactivated pre-existing fault and compare it to the stress needed to form a new fault (Collettini and Sibson, 2001; Mulugeta and Sokoutis, 2003).

The first step is to calculate the optimal fault dip angle $\theta_a$ with respect to the horizontal at which the fault forms during extension to approximate the preferred fault geometry in a first phase of the basin inversion scenario (Equation 4, Figure 2a, after Collettini and Sibson (2001)). This fault angle is defined by the friction $\mu$ of the material so that higher friction leads to steeper faults in the model.

$$\theta_a = 90° - \frac{1}{2} tan^{-1} \frac{1}{\mu} \tag{4}$$

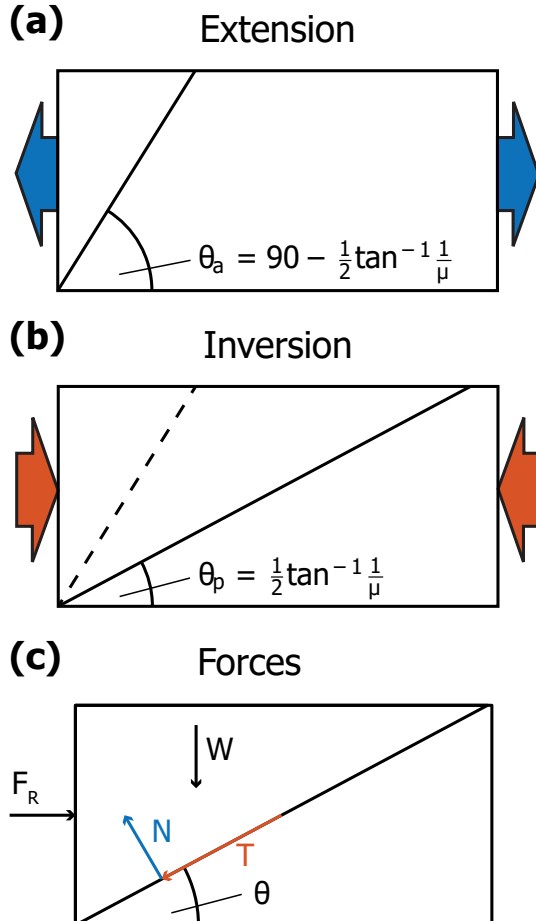

**Figure 2.** Definition of fault angles during a) extension and b) compression. In b) the pre-existing fault is shown as a dashed line showing the difference in orientation. c) Force balance and definition of the force $F_R$ required to move a block of weight $W$ along a fault with an angle of $\theta$ (mod. after Mulugeta and Sokoutis, 2003)

After the extensional phase the basin switches to compression and the optimal angle for a new fault $\theta_p$ with respect to the horizontal is (Figure 2b):

$$\theta_p = \frac{1}{2}tan^{-1}\frac{1}{\mu} \tag{5}$$

To calculate what is energetically more favourable, reactivation of an inherited vs. formation of a new fault, we calculate the horizontal force $F_R$ required to move the wedge of material formed by the side wall, surface and fault angle $\theta$ (Figure 2c). This force additionally incorporates the weight $W = \rho g h$ of the material (Mulugeta and Sokoutis, 2003). For the calculations we assume a normal stress of $\sigma_N = 1000\,Pa$ analogous to the slide-hold-slide tests corresponding to a height $h = 3.5$ to $5.5cm$ for materials with a bulk density of $\rho = 1800$ to $3000\frac{kg}{m^3}$, respectively:

$$F_R = \frac{\rho g h^2 \cot\theta}{2}\left[\frac{\mu + \tan\theta}{1 - \mu\tan\theta}\right] \tag{6}$$

A fault is considered severely misoriented when its angle is twice as large as the optimal fault angle. In this case the term $(1 - \mu\tan\theta) \to 0$ leading to extremely large values for $F_R$. In the locking region $(1 - \mu\tan\theta) < 0$ and therefore $F_R < 0$, which is unrealistic.

To account for healing, the friction for a reactivated fault is time dependent (i.e. increases with hold time) with the healing rate $b$ as the power-law coefficient:

$$\mu_r(t) = \mu_0 t^b \tag{7}$$

This methodology neglects possible edge effects, such as shear stresses along the sidewall, because we assume a continuous granular layer that is cross-cut by several normal faults, which are going to be inverted. Additionally, we do not incorporate changes in the stress field due to differences in elasticity of the material after healing and the formation of lower density shear bands that could lead to stress concentrations. Another important constraint of this simple model is, that it is not suitable for materials with higher cohesion because these tend to form surface cracks during extension leading to a change in fault angle with depth.

All calculations incorporate full uncertainty propagation through the Python module 'uncertainties' assuming normally distributed variables. For the frictional parameters $\mu$, $C$ and $b$ the error given is 2 standard deviations calculated from the covariance of the fit and averaged per material (quartz sand, feldspar sand, glass beads, etc.). For density $\rho$ the value is the arithmetic mean and error is 2 standard deviations of density for each material.

## 3    Results

### 3.1    Grain Characteristics

The materials are well sorted and very homogeneous because they are standardised industrial products for specific purposes. As a consequence, they contain no to few impurities, such as clay or pebbles, and are mostly monomineralic. Therefore, the properties presented here should apply to all batches from the same manufacturer. See Table A1 for a more detailed description of each sample and Figure 7 for a comparison of quality index with the frictional properties.

#### 3.1.1    Quartz Sands

Quartz sands are the most frequently used analogue material and therefore represent the majority of samples. The color of most sands is yellowish to white with clear to translucent grains. Some sands are very homogeneous consisting of more than 99 % quartz while others contain considerable (>5 %) traces of feldspars, mica and other minerals. The sphericity is medium to high across all samples and differences are only minor. Roundness shows a larger spread with some rounded samples, such as the GFZ sands (Figure 3g), and some very angular sands, e.g. from Wroclaw. Roundness is mostly derived from the origin of the sands. Some sands are unprocessed eolian or fluvial sands that are (sub)round, while others have been processed and therefore have high angularity due to crushing (Figure 3j). This division is also evident in the large spread of surface roughness. The sands that are rounded to subrounded generally have smoother surfaces. The angular sands often have shelly or jagged surfaces leading to high surface roughness. Some sands seemingly are mixtures of rounded and angular sands and therefore receive a lower score.

#### 3.1.2    Glass Beads

Most glass beads show a very high sphericity, are perfectly rounded and have very smooth surfaces. Depending on the manufacturer, they are very well sorted and only have very few impurities (Figure 3e), e.g. glass beads from GFZ and Prague that are both supplied from the same manufacturer. Some samples either contain non-spherical grains (Figure 3f), fragments or a significant amount of beads that are sticking together or have small protrusions. This leads to a slightly lower score for sphericity and roundness. Usually, the glass beads are perfectly clear with a few whitish dots on the surface that probably stem from impacts of other beads during manufacture and transport.

#### 3.1.3    Corundum Sands

The corundum sands, which are exclusively processed (crushed) material, have medium sphericity with some elongated grains leading to a devaluation to a lower score in comparison to quartz sands. The color is yellowish, brown but transparent for the brown corundum sands and clear for white corundum. In general, they are very angular with sharp and pointy edges (Figure 3k). This also leads to a high surface roughness, although the faces between the edges are generally flat and smooth. Some surfaces are shelly adding even more surface roughness. As a result, the average quality score is very low.

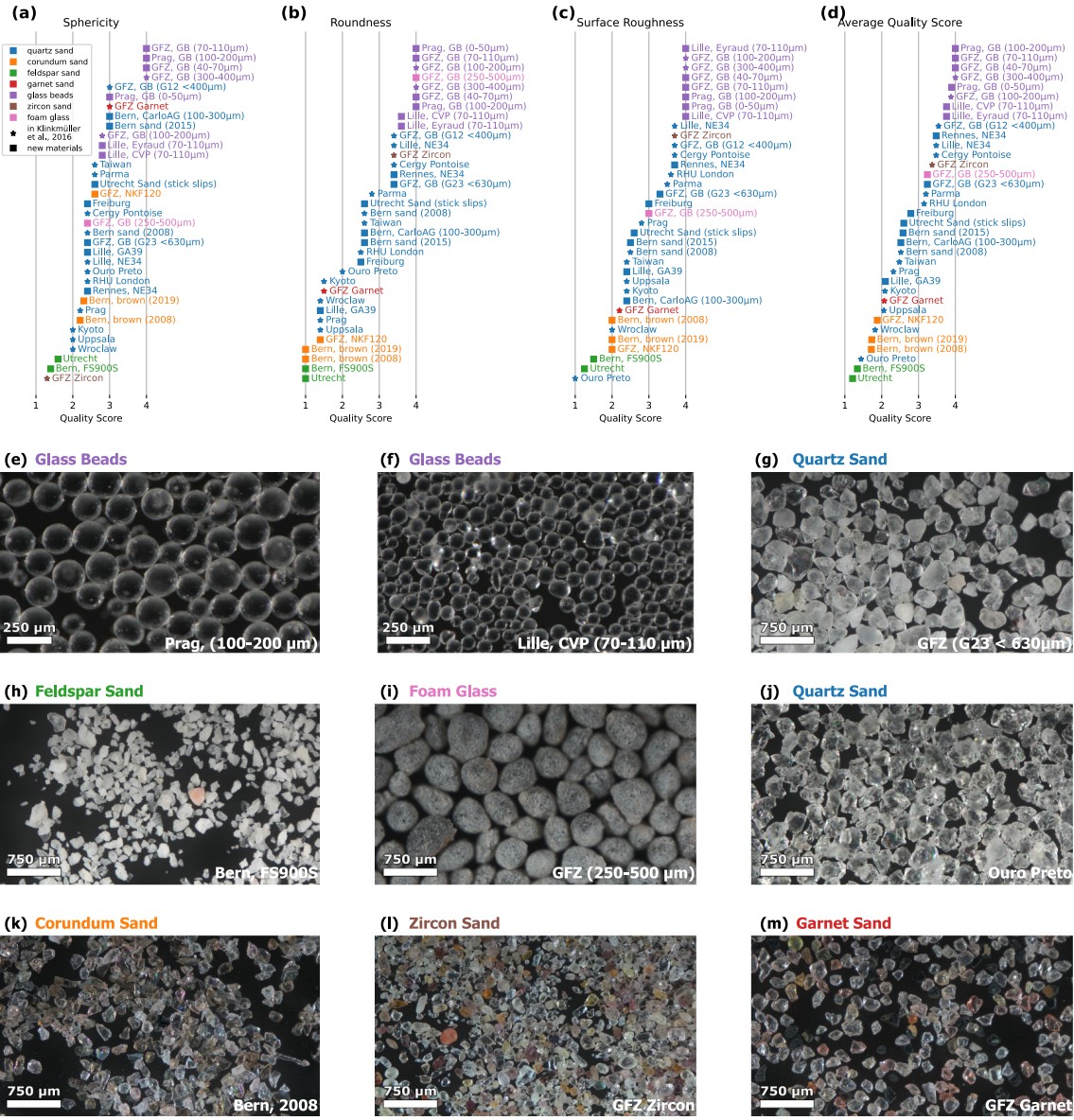

**Figure 3.** Quality scores and some exemplary materials that show specific characteristics. a)-d) Stacked and sorted quality score for the individual properties and weighted average according to Table 2. e) Glass beads with high sphericity, high roundness and low surface roughness. f) Glass beads with some low-sphericity grains. g) Typical quartz sand with medium sphericity, good roundness and low surface roughness (eolian sand). h) Feldspar sand with low sphericity, low roundness and high surface roughness. i) Foam glass with medium sphericity, good roundness and medium surface roughness. j) Sand with high surface roughness due to shelly and jagged surfaces (crushed sand). k) Corundum sand with some elongated grains leading to a higher sphericity score. l) Zircon sand with a many elongated and elliptical grains but with good roundness. m) Garnet sand with spherical, sub-angular grains and high surface roughness due to broken grains.

### 3.1.4 Feldspar Sands

The feldspar sands feature milky to translucent, white grains with low sphericity due to their elongated and triangular shape (Figure 3h). They are very angular and some seem to be aggregates of smaller grains. The surfaces are very rough with many sharp edges and surfaces, possibly due to cleavages. Most grains are internally fractured, which contributes to the milky and translucent appearance. Therefore, the feldspar sands have the lowest average quality score.

### 3.1.5 Zircon Sands

The zircon sand, which is like corundum sands crushed, is poorly sorted and contains a large variety of grain sizes and grain shapes (Figure 3l). Many grains are elongated and show the characteristic habit of zircon crystals, some seem to be fragments of these larger crystals. The color is reddish brown to off-white and the grains are translucent to clear. Depending on the grain size the grains are angular to rounded. Large grains tend to be well rounded and almost spherical while smaller ones are angular to sub-angular. The more or less intact crystals have sub-rounded crystal faces and edges. Due to the heterogeneous composition, the surface roughness also has a strong variation. The majority of grains has a smooth surface, however there are a few that show shelly or slightly rough surfaces.

### 3.1.6 Garnet Sands

On average, the garnet sands have spherical grains with a small proportion of elongated grains (Figure 3m). They are mostly transparent with a reddish tint and about 10 % of grains are dark and opaque. The transparent grains are angular to sub-angular and have a shelly surface. The darker grains are sub-rounded and have a slightly rough surface. Additionally, larger grains seem to be more rounded than smaller ones.

### 3.1.7 Foam Glass

Similar to the glass beads, the foam glasses are an industrially manufactured product and therefore are very homogeneous. No impurities could be found and the grain size distribution is very narrow and corresponds to the given specifications. They have a medium sphericity and have a ellipsoidal to random shape, similar to asteroids (Figure 3i). The color is gray, and the grains are opaque. The grains are well rounded with no visible edges or faces. They have a very fine surface, similar to sandpaper, resulting in a slightly rough surface. Due to their surface roughness they rank just below the glass beads but still above the quartz sands.

## 3.2 Healing and Compaction Rates

We find that most materials exhibit healing rates in the range of $b <= 0.03$ (Figure 4), i.e., an increase in strength of less than 3 % per tenfold increase in hold time (e.g. 10 s vs 100 s). Only two samples show healing rates that are not significantly different from zero, 'Freiburg' and 'Bern, CarloAG (100-300 $\mu m$)' and are therefore considered to not show healing over experimental time scales. Most quartz sands show lower than average healing rates of roughly 1 % per tenfold increase in hold

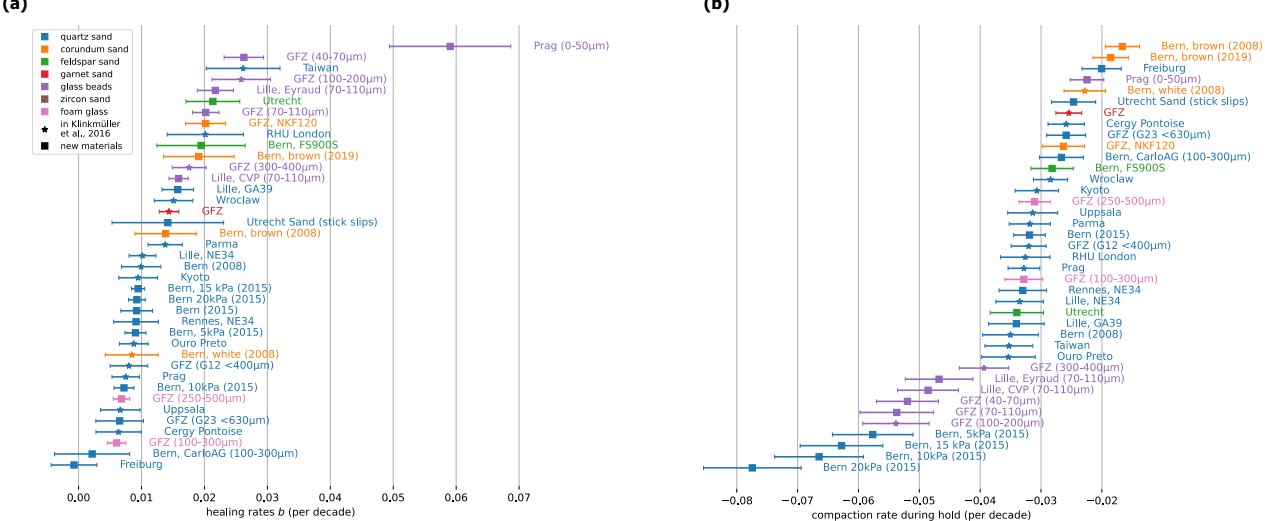

**Figure 4.** Stacked and sorted plot of (a) healing rates and (b) compaction rates for all tested materials. The numbers either indicate grain size range (XX-YY $\mu m$), type (e.g., G23) or year of acquisition (20XX). Similar materials often plot in the same range of healing or compaction rate. Notable exceptions are the healing rate for very fine glass beads (Prague, 0-50 $\mu m$) and the compaction rates for higher normal stresses (Bern, 5 kPa - 20 kPa).

time. Foam glass shows healing rates generally lower than those of quartz sands. Glass beads exhibit higher healing rates and show considerable spread between $b = 0.015$ and $0.025$. Corundum sands are distributed over the full range of healing rates and feldspar sands overlap with glass beads. However, the number of samples for materials other than quartz sands and glass beads, is too small to generalise.

Compaction rates are more diverse and show a clearer separation between the materials (Figure 4b). Glass beads show the strongest compaction rates of $c = -0.055$ to $-0.04$ at the reference normal stress of $\sigma_N = 1\,kPa$. This means that sample thickness decreases by 4 to 5.5 % per tenfold increase in hold time. The only exception are the very fine glass beads (Prague, 0-50 $\mu m$), which show a very low compaction in comparison to the other glass bead samples. Quartz sands show small compaction rates between $c = -0.04$ and $-0.02$, with the majority of quartz sands having $c <= -0.03$. Feldspar sands and foam glass compaction rates are again comparable with quartz sands. In general, corundum sands exhibit the lowest compaction rates.

There is a strong influence of normal stresses on compaction, which leads to much stronger compaction during hold. However, the healing rate is not significantly correlated with higher compaction and therefore the measurements with higher normal stresses form a distinct cluster when plotting compaction rate versus healing rate (Figure 5). Most samples do not show a clear distinction between sample material, compaction and healing rate. They all plot in a single cluster, with no significant correlation of compaction rate and healing rate. The glass beads are exceptional because they show high healing rates and high compaction rates, therefore forming a separate cluster. Using main and glass bead cluster, a weak negative correlation

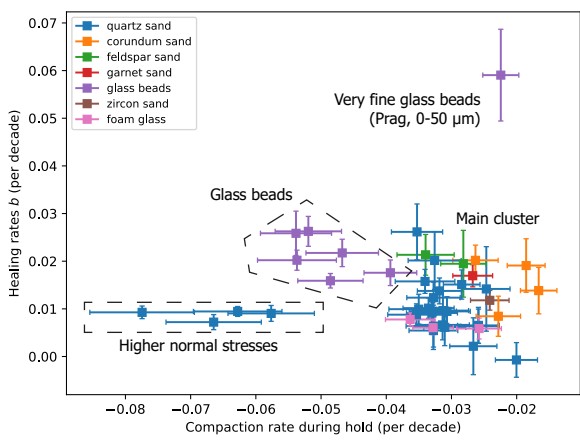

**Figure 5.** Comparison of healing rate and compaction rate. Materials that behave similarly are grouped in clusters.

of healing rate and compaction rate is visible. An increase of healing with stronger compaction might be recognizable, albeit not statistically significant. The most notable exception are again the very fine glass beads (Prague, 0-50 $\mu m$). These show a
significantly stronger healing at small compaction rates.

### 3.3  Reactivation Properties

For the reactivation properties we summarised the samples into seven groups by taking the average and the standard deviation $2\sigma$ of all properties (friction coefficients $\mu$, healing rate $b$ and density $\rho$). As material height we choose $h = 0.05\,m$ that is comparable to a normal stress of $\sigma_N = 1000\,Pa$ for most materials and lies in the range of typical analogue model setups.
For the first phase of a typical basin inversion model the material is subjected to extensional stresses ($\sigma_1 \parallel S_v$, Equation 4) and fault angles are defined by the materials peak friction of $\mu_p = 0.50$ to $0.75$, which results in $\theta_a = 58$ to $64°$ (dashed, coloured lines in Figure 6a-g). During compression the stress field changes ($\sigma_1 \parallel S_{hmax}$) and so does the angle of newly generated faults (Equation 5). The friction coefficient is the same because new faults mainly cross-cut undisturbed material and angles are in the range of $\theta_p = 26$ to $32°$, which leads to flatter faults in the model (solid, coloured lines in Figure 6a-g).
The locking regions of all materials become increasingly larger, as reactivation friction $\mu_r$ increases over time. We find that for all materials, with the exception of glass beads, the angles of the pre-existing faults fall within the locking region. As a result, none of the faults that were created during extension should reactivate because they are severely misoriented. The optimal angle for reactivation using the time-dependent $\mu_r$ is similar to the optimal angle of a new fault with $\mu_p$. This means that the difference in friction due to healing is not large enough to facilitate sliding along the inherited normal faults.
We find the same for the force required to move a wedge of material along the pre-existing fault or the creation of a new fault (Figure 6h-n). For new faults the shear force per unit area ranges between $F_R = 50$ and $150\,\frac{N}{m}$, which means that for a triangular wedge of 5 cm height and 100 cm length in the direction of $\sigma_2$ between 50 and 150 N are required to initiate a new

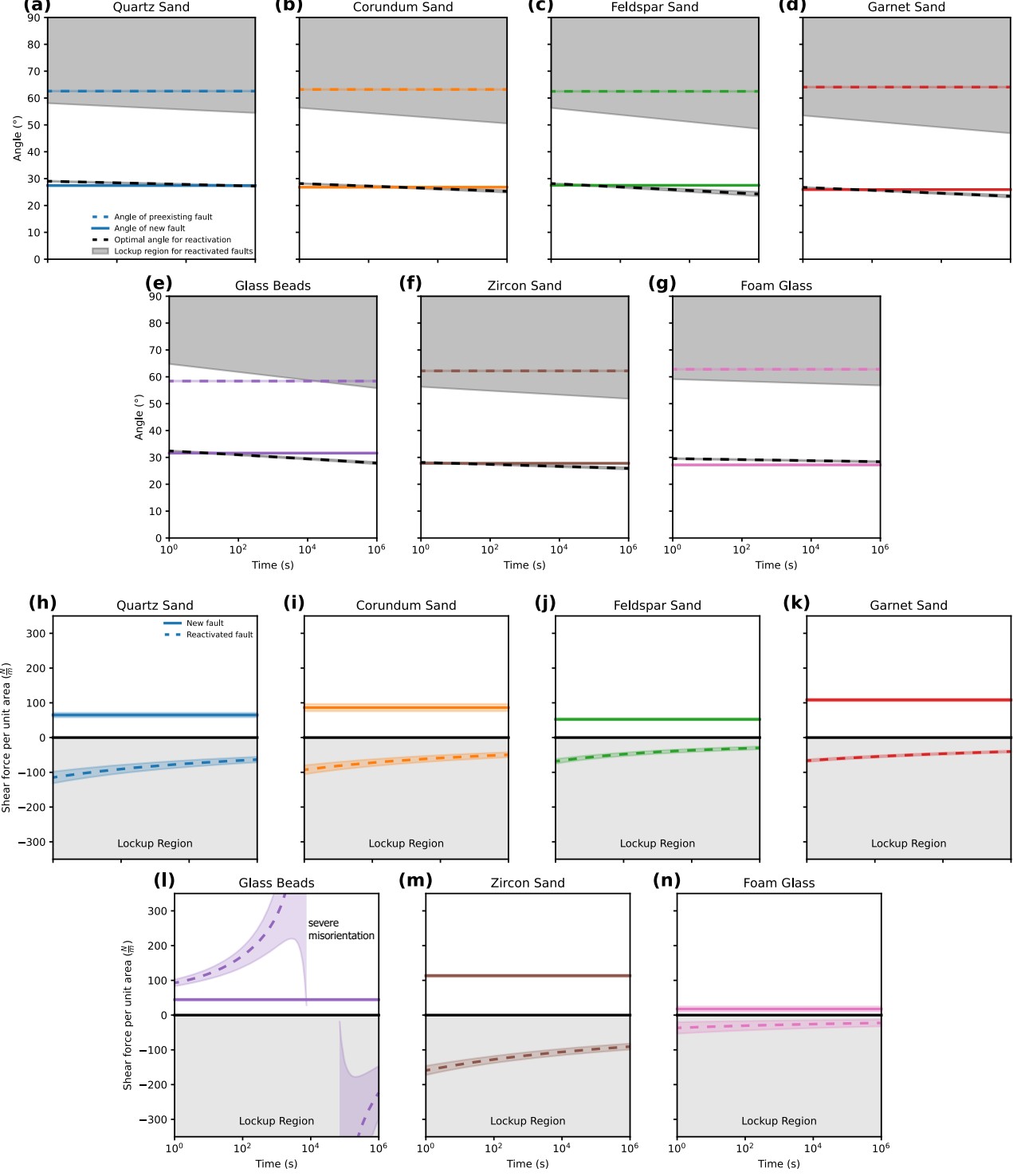

**Figure 6.** Optimal angles and shear forces for new and reactivated faults. a)-g) Optimal angles of faults for each material. h)-n) Shear forces required to move a wedge of material along a new and reactivated fault. Errors are derived from the uncertainties of the parameters and represent $2\sigma$.

fault. While the height of the wedge has an influence on the absolute values of forces, the ratio between the forces $\frac{F_{new}}{F_{inherited}}$ is independent of height. For reactivated faults most materials show negative values and therefore reactivation is not possible.

The only exception are glass beads, which still are in the field of possible reactivation up to $t_h = 10^4\,s$ after shear. However, the stress required to reactivate these faults roughly is twice as high as for creating a new fault and reaches extremely high values already after $t_h = 10^3\,s$ (Figure 6l). A close inspection of the individual glass bead samples shows that glass beads with low grain sizes (e.g. GFZ, GB 70-110 $\mu m$) are fully outside the locking region. The stresses to reactivate these faults are only 1.5 to 2 times higher than the stress for new faults. The fault orientation is still more than 20 degrees away from the optimal

but under certain conditions a reactivation is possible for this sample. In general, reactivation is very unlikely despite the small difference between optimal and inherited angles.

## 4    Discussion

### 4.1    Impact of Time Consolidation

#### 4.1.1    Reactivation of Faults in Basin Inversion Models

Our results show that for the tested materials the reactivation of inherited normal faults generated during extension as reverse faults should generally not be possible. This means that the difference between peak and reactivation friction of 5 to 15 % is too small to create faults outside the locking region (Sibson, 1980). Additionally, extensional faults in typical analogue models show higher angles than what is inferred from friction measurements with ring-shear testers (Panien et al., 2006). This is in accordance with the results of other studies that show only weak to no reactivation of pre-existing structures in purely frictional

sandbox models (Jara et al., 2018; Marques and Nogueira, 2008; Almilibia et al., 2005; Yagupsky et al., 2008; Molnar and Buiter, 2022). Due to their relatively low friction coefficient, glass beads are a certain exception. Reactivation is however still unlikely due to the high stresses required. As a consequence, many modellers use strong boundary conditions, such as heterogeneities or blocks, to force fault reactivation along specific normal faults (Bonini et al., 2000).

     For setups that aim to reactivate extensional structures that are generated in situ, a possible solution could be to increase

the amount of extension. On average the misorientation between locking region and fault orientation is less than 10°. During extension the blocks rotate in a bookshelf like motion, thereby creating flatter fault angles. The stresses to reactivate faults close to the locking regions are still quite high and therefore a rotation of $\approx 20°$ or more is required to lead to structures that can be reactivated. In analogue models the fault orientations usually show a larger spread than what is calculated in our theoretical model. The graben systems that form during extension usually contain several fault angles with flatter and steeper segments.

Therefore, some faults could already be well within the reactivation field due to the heterogeneity in the analogue model, e.g. due to variation in bulk density.

     Our model has limitations in quantifying the effect of cohesion on the reactivation of faults. The ring shear tests suggest that the reactivation cohesion is up to twice as high as peak cohesion for most materials, which hinders fault reactivation even more. This effect is amplified for models with small thicknesses because for these the ratio of cohesive to gravitational stress

is larger (Ritter et al., 2016a). To decrease the effect of cohesion the only option is to increase the normal stress by increasing the layer thickness. We note however, that cohesion is not directly measured here but is inferred from extrapolation. Because cohesion in granular materials used in analogue modelling is typically very small (few tens of Pa) or even zero, differences in cohesion might not be quantitatively sound.

### 4.1.2 Implication of Healing Rates on Basin Inversion Models

Regardless of the mechanism creating the faults, most materials show a measurable amount of healing. The healing rates in the tested materials are low to moderate and lead to an increasingly larger locking region over time. For materials that have high healing rates, e.g. glass beads, this increase is up to 1.5° per tenfold increase in hold time. This means that granular faults that are about 12 hours old require a 7° lower angle to remain in the activity field. Depending on the type of setup this can change the number of active faults, especially in graben systems consisting of several, similarly oriented faults. Healing can lead to a

smaller amount of active faults when the time between fault creation and fault motion is increasing. In theory this should also apply to precut faults, because they alter the grain packaging in a similar way. This could even lead to the creation of new faults instead of localization at the predefined locations. However, with our methodology this is not verifiable and other mechanisms, such as the formation of shear fabrics in the granular material could play a role.

Due to the exponential nature of Equation 7 healing is very strong immediately after a fault comes to rest. Changing stress

distribution in a running model commonly leads to constant, simultaneous activity on many faults. Therefore, only when a model is stopped, i.e, the model settles under the influence of gravity, healing can have a visible effect. Healing rates are low for most materials, meaning that no effect on reactivation angles can be expected for several hours, although healing is strongest in the first few hours. In comparison with other processes, such as sieving technique or boundary conditions, fault healing only has a small, secondary influence on fault behaviour like the velocity dependence of friction during shear (Rosenau

and Oncken, 2009). Consequently, if a model is continuously run without longer interruptions ($t_h < 1\,h$), the effect of healing is indistinguishable from other instantaneous mechanisms that influence an analogue fault's strength.

### 4.1.3 General Impact of Time Consolidation on Analogue Models

In the general context of analogue models, time dependent change of fault strength has further effects whose influence is small but should nevertheless be taken into account. There is a class of analogue models, also common in basin inversion, where

frequent interrupts are inevitable. Typical examples are models with concurrent sedimentation or erosion where material is added or removed during a hold phase (e.g., Bonnet et al., 2007; Graveleau and Dominguez, 2008; Molnar and Buiter, 2022). With the increased use of computer tomography or laser based topography scanning to screen experiments comes another type of model that must be interrupted at regular intervals. As mentioned in section 4.1.2, interruptions that are shorter than one hour should not have a great effect. The change in normal stress due to the addition or removal of overburden should have a

much stronger effect than the restrengthening due to time consolidation. During the addition of material and the associated preparation of the model surface, a hardening of the fault zone could occur. For example, smoothing the surface with a spatula

or ruler could exert additional stress on the fault zone and thus promote compaction. However, this effect is already commonly known for such models and has a much greater impact than the healing.

Repeatability of models can become an issue when the timing of the individual model runs is not the same. In recent years, more attention has been paid to the repeatability of analogue experiments. For example, Santimano et al. (2015) were able to demonstrate that various parameters such as fault length and activity are intrinsically variable. Experiments where fault activity is monitored with or without erosion therefore could be influenced and even altered by repeated interruption (Hoth et al., 2006, 2007). Likewise, in many studies, a repetition of simple experiments is carried out to quantify the influence of preparation, setup and climatic conditions. Time strengthening brings another factor into play here. For example, models that have the same setup, but run for different lengths of time due to the duration, could have differences in fault activity or fault orientation. An experiment that is left overnight or for a longer period of time tends to have a higher strength than a model that is deformed within a short period of time. There is anecdotal evidence of analogue models that have failed to reach the expected results when the material has been sitting in the box for too long. For example, a typical sandbox experiment is performed more frequently at GFZ Potsdam's analogue Laboratory (HelTec) for demonstration purposes. It was found that for a representative result, the time difference between preparation and execution should not exceed one day. Therefore, especially in experiments with heterogeneities, for example due to faults or precut surfaces, care should be taken that the duration of inactivity of the faults is approximately the same in repeated experiments.

## 4.2 Relation of Frictional Properties with Grain Characteristics

We compare the quality score of each material with all frictional parameters to recognise possible influences. To quantify correlations we calculate Pearson $r$ and Spearman $\rho$ correlation coefficients. The Pearson correlation coefficient assumes a linear correlation between two variables, but this is not always the case. Therefore, we also use the Spearman correlation coefficient that is independent of the underlying distribution. For both parameters the p-value is determined, which gives an indication of the significance of correlation. Generally, if $p < 0.05$ one assumes that the found correlation is not possible through random permutation of the original data and therefore statistically significant. It should be noted, however, that in this manuscript the quantification of the quality score is subjective, while the friction parameters including their errors are based on real measured values. Thus, the use of correlation coefficients is only an aid to identify trends more easily in the set of measurements and samples.

We find that the healing rate $b$ does show a weak positive correlation with sphericity ($r = 0.39$, $p = 0.05$; $\rho = 0.40$, $p = 0.04$, Figure 7a) and is otherwise not correlated with any other quality measure. This means that with higher sphericity the materials tend to have higher healing rates. This is probably related to the compaction rate, which shows a similar negative correlation with sphericity. Spherical grains compact more than aspherical beads, as indicated by the higher compaction rates of glass beads (Figure 5 and Figure 7e). This is consistent with a tendency of glass beads to compact more easily during cyclic axial loading tests reported by Klinkmüller et al. (2016). Higher sphericity is associated with lower void ratios before and after shearing (Härtl and Ooi, 2011) and therefore the individual grains are compacted more during a hold phase. In consequence, the stress required to mobilise these materials is higher because a larger amount of dilation is needed.

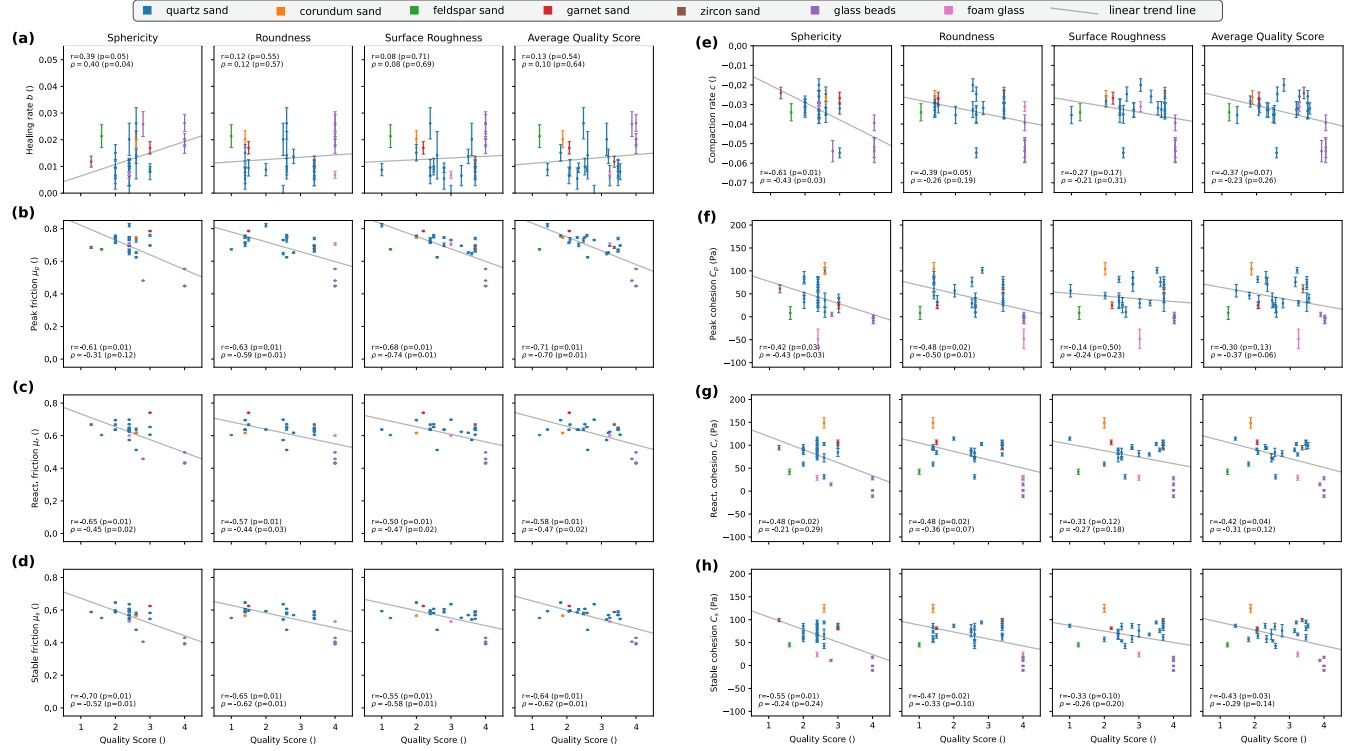

**Figure 7.** Correlation of frictional properties with quality score. The grey lines are simple linear fits of the data to highlight the trends in the datasets. They are not necessarily of statistical significance. a) Healing rate, b-d) Friction coefficients, e) Compaction rate), f-h) Cohesions.

All three friction types (peak, reactivation and static, Figure 7b-d) show a moderate negative correlation with all quality scores ($r \approx \rho \approx -0.6$, $p < 0.05$). As a result, there is a tendency to get lower frictional coefficients for materials that have high sphericity, higher roundness and smooth surfaces. The negative correlation of quality index with cohesion is very weak and has a much higher p-value leading to a higher uncertainty in the correlation (Figure 7f-h). These results are in accordance with previous studies on granular characteristics (Klinkmüller et al. (2016); Panien et al. (2006) and references in Table 2). We find a high correlation of peak friction with average quality score ($r = -0.70$, $\rho = -0.67$, $p = 0.01$) mostly influenced by the correlation of peak friction with roundness and surface quality.

The exceptionally high healing rate for very fine glass beads (Prague, 0-50 $\mu m$) is probably the result of a strong increase in cohesion. In contrast to the other glass bead samples they do only show a small compaction rate, i.e., there is only a small reduction of density. Because of the wide particle size distribution and large amount of fine materials the void space is largely filled with finer glass powder. Additionally, very fine granular materials are susceptible to electrostatic effects that increase the attractive forces between particles.

Another outlier is the sand from Utrecht that shows stick-slip. Due to the stick-slip behaviour, we assume an increased healing rate, as in this case there can theoretically be several effects. The higher the healing rate $b$ the greater the difference

$a - b$ with the direct effect $a$ being constant. In theory, this makes the material more velocity weakening and thus more unstable (Dieterich, 2007). Additionally, with higher $b$ the critical stiffness increases. Therefore, a system that was previously stable, such as the ring shear tester, now needs to have a higher stiffness to sustain stable sliding. As a result the system shows a higher tendency for stick-slips. However, due to the stick-slips it is not possible to measure the healing rates adequately. This would require a much higher apparatus stiffness which was not possible to change for our ring shear tester.

## 4.3   Suitability for Modelling Natural Faults

Geodetic studies on postseismic surface motion demonstrate that faults quickly relock after a large earthquake (Bedford et al., 2016). A similarly high post-slip healing rate is required to explain the observations in analogue models of subduction zone kinematics using sticky rice as a fault zone analogue (Rosenau et al., 2009; Kosari et al., 2022). We observe the same strong healing in the first few hold intervals caused by the power-law relation of strength and hold time (Equation 7), which is also observed for laboratory experiments on synthetic and natural fault rocks (Karner et al., 1997; Carpenter et al., 2016; Scuderi et al., 2014). Healing rates for dry synthetic and natural fault gouges range between 0.001 and 0.01 at room temperature (Ikari et al., 2016). Under wet conditions this healing rate can increase by one order of magnitude at temperatures $> 300\,^{\circ}C$ (Niemeijer et al., 2008, and references therein). For gouges containing salt or salt-muscovite layers the rates are even higher. A general correlation between the healing properties and the behaviour of fault zones thus seems plausible.

Quartz sands and foam glass show healing rates that are generally low and closer to dry gouges while the glass beads, especially the very fine glass beads, have values that are closer to hydrothermal healing rates. However, the frictional strength is too low to use glass beads as well scaled analogues of self-healing faults in a model. A possibility might be the addition of a small percentage of fine material to sands thus achieving cohesion driven healing while retaining the overall higher bulk strength of sand. Such material mixes were not part of this study but can easily be tested with the methodology outlined here.

## 5   Conclusions

Healing rates and grain characteristics of a set of commonly used analogue materials from different laboratories were acquired through slide-hold-slide tests and by qualitative description. Furthermore, the reactivation potential of inherited normal faults in these materials was calculated using additional information from previous ring-shear tests. These experiments provide a better understanding of transient, time-dependent changes in analogue models and their potential impacts on the suitability of these materials for certain types of analogue models, such as basin inversion models. The experiments show that:

1. There is a measurable time dependency in brittle analogue models in the form of fault strengthening over the experimental timescale.

2. The time consolidation is small in comparison to other factors that influence the strength of analogue materials, such as preparation technique or climatic conditions, i.e., air humidity. However, there is a possible negative impact on re-

peatability and model outcome for long-running experiments or experiments with substantial downtime between the deformation phases.

3. Healing rates are generally low but comparable to natural faults and gouges. The use of the tested analogue materials as analogues of healing faults might be possible. Glass beads show the highest amount of healing due to their spherical shape and smooth surface, which leads to increased compaction during hold.

4. Reactivation of pre-existing faults in the tested granular materials is very unlikely if the faults are not manually predefined. Fault orientations generated by extension are too steep and always lie in the locking region. Only overextending with associated block rotation could lead to faults that are in the reactivation region. However, the generation of new faults is almost always energetically more favourable.

5. The frictional properties of most materials only show a weak correlation with grain characteristics. The strongest corre-
lation was found for healing rates with sphericity and friction with average quality score. A general trend is that a low quality score roughly correlates with higher friction.

*Code and data availability.* All data is going to be available in the form of a data publication (Rudolf et al., 2023), together with Python scripts to generate all figures that use data (Rudolf, 2023). The python package 'granular-healing' for the analysis of slide-hold-slide tests is open-source and is part of the data publication as well as in a public git repository (https://git.gfz-potsdam.de/analab-code/granular-healing).

*Video supplement.* A video showing the experimental procedure of the ring-shear test is going to be available on the EPOS channel on youtube.com upon the publication of this manuscript.

**Table A1.** Description and quality score for each sample. Abbreviations: Qz - quartz, Crn - corundum, Fsp - feldspar, GB - glass beads, Grt - garnet, Zrn - zircon.

| No. | Name | Sphericity Description | Score | Roundness Description | Score | Surface Description | Score |
|---|---|---|---|---|---|---|---|
| 02 | Qz sand, Bern Neusand | high | 2 | subrounded to subangular | 2.4 | slightly rough, shelly, some jagged | 2.5 |
| 03 | Qz sand, Utrecht 2018 | high to medium | 2.4 | subrounded to subangular | 2.4 | sightly rough, shelly | 2.4 |
| 05 | Qz sand, Rennes NE34 2021 | medium to high | 2.6 | subrounded to rounded | 1.6 | smooth, few shelly, few slightly rough | 1.3 |
| 06 | Qz sand, Parma | high to medium | 2.4 | subrounded, few subangular | 2.2 | smooth, shelly | 1.5 |
| 07 | Qz sand, Kyoto | medium | 3 | angular to subangular, few subrounded | 3.5 | shelly, slightly rough | 2.6 |
| 08 | Qz sand, Bern 2008 | medium to high | 2.6 | subrounded to subangular | 2.4 | slightly rough, shelly, some jagged | 2.5 |
| 09 | Crn sand, GFZ NKF120 | high to medium | 2.4 | angular to subangular | 3.6 | shelly | 3 |
| 10 | Qz sand, Uppsala | medium | 3 | angular to subangular | 3.6 | shelly, slightly rough | 2.6 |
| 11 | Qz sand, Cergy Pontoise | medium to high | 2.6 | subrounded to rounded | 1.6 | smooth, few shelly, few slightly rough | 1.3 |
| 12 | Crn sand, Bern 2019 | medium to high, few elongated grains | 2.7 | angular | 4 | shelly | 3 |
| 13 | Fsp sand, Utrecht | medium to low | 3.4 | angular | 4 | jagged, rough | 3.75 |
| 14 | Qz sand, RHU-London | medium to high | 2.6 | subangular, some angular, some rounded | 2.5 | smooth, some shelly | 1.4 |
| 15 | Qz sand, GFZ G12 $<400\,\mu m$ | high | 2 | subrounded to rounded | 1.6 | smooth, few shelly, few slightly rough | 1.3 |
| 16 | Qz sand, GFZ G23 $<630\,\mu m$ | medium to high | 2.6 | subrounded to rounded | 1.6 | slightly rough, smooth, few shelly | 1.7 |
| 17 | Qz sand, Prag ST55 | medium | 3 | angular to subangular | 3.6 | slightly rough, few shelly | 2.2 |
| 18 | Grt sand, GFZ 2008 | high | 2 | angular to sub angular, few rounded | 3.5 | shelly, few slightly rough to smooth | 2.8 |
| 19 | GB 70-110 $\mu m$, GFZ | perfect | 1 | rounded | 1 | smooth | 1 |
| 20 | GB 300-400 $\mu m$, GFZ | perfect | 1 | rounded | 1 | smooth | 1 |
| 21 | GB 40-70 $\mu m$, GFZ | perfect | 1 | rounded | 1 | smooth | 1 |
| 22 | GB 100-200 $\mu m$, GFZ | high, some medium | 2.2 | rounded | 1 | smooth | 1 |
| 23 | Zrn sand, GFZ 2009 | low to medium, several elongated grains | 3.7 | subrounded to rounded | 1.6 | smooth, few shelly, few slightly rough | 1.3 |
| 25 | GB 0-50 $\mu m$, Prag poured | high | 2 | rounded | 1 | smooth | 1 |
| 26 | GB 100-200 $\mu m$, Prag | perfect | 1 | rounded | 1 | smooth | 1 |
| 28 | Qz sand, Bern CarloAG 100-300 $\mu m$ | high | 2 | subrounded to subangular | 2.4 | shelly, slightly rough | 2.6 |
| 35 | Fsp sand, Bern FS900S | low to medium | 3.6 | angular | 4 | rough, jagged | 3.5 |
| 36 | Qz sand, Lille G39 | medium to high | 2.6 | angular to subangular | 3.6 | shelly, slightly rough | 2.6 |
| 37 | Qz sand, Lille NE34 | medium to high | 2.6 | subrounded to rounded | 1.6 | smooth, few shelly, few slightly rough | 1.3 |
| 38 | GB 70-110 $\mu m$, Lille CVP | high, some medium | 2.2 | rounded to subrounded | 1.4 | smooth | 1 |
| 39 | GB 70-110 $\mu m$, Lille Eyraud | high, some medium | 2.2 | rounded to subrounded | 1.4 | smooth | 1 |
| 40 | Qz sand, Prague | medium to high | 2.6 | angular to subangular | 3.6 | slightly rough, few shelly | 2.2 |
| 41 | Qz sand, Wroclaw | medium | 3 | angular to subangular | 3.6 | shelly | 3 |
| 42 | Qz sand, Taiwan | high to medium | 2.4 | subrounded to subangular | 2.4 | shelly, slightly rough | 2.6 |
| 43 | Qz sand, OuroPreto | medium to high | 2.6 | subangular | 3 | jagged | 4 |
| 45 | Crn sand, Bern | medium to high, some elongated grains | 2.8 | angular | 4 | shelly | 3 |
| 46 | Qz sand, Freiburg | medium to high | 2.6 | subangular to round | 2.5 | some smooth, some jagged | 2 |
| 47 | Foam glass 250-500 $\mu m$, GFZ | medium to high | 2.6 | rounded | 1 | slightly rough | 2 |

*Author contributions.* Conceptualization of this project was done by MRu and MRo. Measurements were done by MRu and MRo. Data analysis and programming was done by MRu. The manuscript was written by MRu with the help of MRo and OO. All authors read and approved the final manuscript.

*Competing interests.* The authors declare that no competing interests are present.

*Acknowledgements.* We thank all participants from the laboratories who sent their samples to be tested for this study. The authors thank the technical staff at the Helmholtz Laboratory for Tectonic Modelling for running some of the long-term tests (F. Neumann) and for setting up the monitoring devices (T. Ziegenhagen). We kindly thank J. Mingram (GFZ Potsdam) and L. Stutenbecker (TU Darmstadt) for providing support and access to the microscopes at their institutes. In addition, we thank H. Elston and an anonymous reviewer for their helpful
comments and ideas that have been instrumental in moving this manuscript forward. This research has been partially funded by Deutsche Forschungsgemeinschaft (DFG) through grant number 235221301 - CRC 1114 "Scaling Cascades in Complex Systems", Project B01 "Fault networks and scaling properties of deformation accumulation".

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
