# Peer review of "Time-Dependent Frictional Properties Of Granular Materials Used In Analogue Modelling: Implications For Mimicking Fault Healing During Reactivation And Inversion"

_EGUsphere, 2022_

## Author Response (AR2)

**Responses to Reviewer Comments**

We thank the two reviewers for their very helpful comments which helped to improve this manuscript. Some comments and questions are very similar, therefore we combined them where we saw fit.

**New tests or old tests?**

**Reviewer 1:** "Whether ring-shear tests were conducted for this study specifically or results from previous studies were reanalyzed for this study is not entirely clear. Lines 73-76 insinuate that the authors collected samples of 14 granular materials and conducted ring-shear and slide-hold-slide tests, however, lines 121-122 indicate that the ring-shear test data is taken from previous studies and reanalyzed for this study. I believe the results are from a combination of new and previous test, but a sentence either in the last paragraph of the introduction or early on in the methodology would clear up any potential confusion."

**Reviewer 2:** "The paper is not clear about whether new tests were performed for this study or results from previous studies were used. Line 73-75 seems to state that new tests were conducted, but then lines 121-123 indicate that data were taken from previous studies and reanalysed. I believe you have conducted new measurements and reanalysed previous data, but this is an important point that should be clarified. In addition, Table 1 includes multiple references, but the legend in figure 3 only highlights data from Klinkmüller et al. (2016). Does this mean that data from the other papers cited in this table were not considered in this study?"

We used old materials that were archived at our laboratory (e.g. from Klinkmüller et al., 2016) and new materials that we got sent from several laboratories. For most of the old samples we had existing 'default' measurements for the basic frictional properties (Mohr-Coulomb strength, peak, static and dynamic friction coefficients and cohesion). We reanalysed these older datasets with our new refined software which is now our defacto laboratory standard workflow. For materials where we did not have previous measurements or where the data was incomplete we performed new ring shear tests for these properties. For *all* materials we performed the slide-hold-slide tests to get the healing rates. These experiments were not done beforehand, only for a very small part of the materials (Rudolf et al., 2021) and not for such long hold times. To clarify this issue we added another subsection 'Materials'. In addition, we have modified Table 1 to list all samples individually with their respective publications and which new measurements were made as part of this manuscript.

**Definition of Quality Scores**

**Reviewer 1:** "The method to assign grain characteristics or quality scores and associated uncertainties is unclear unless reading the background literature. A brief explanation of how quality scores are assigned and how precise the quality scores are would be helpful, especially when interpreting Figure A1 and the calculated correlation coefficients. Additionally, including some description of the correlation coefficients and comparisons between the quality scores and healing rates, friction coefficients, and cohesion would strengthen the methodology section."

**Reviewer 2:** "It is not clear just by reading the paper how quality scores and weights were assigned and I think a brief explanation needs to be provided, especially to interpret Figure A1. In addition, I think Figure A1 should be moved to the main text because it shows correlations that are discussed in the section 4.2."

We added a more detailed description on how the quality score is determined to the Section "Grain Characteristics". Table 2 now contains schematic images of typical grains to illustrate how the parameters are defined. Furthermore, we added a paragraph on the motivation behind the quality score. It should be noted that the quality score is not a defined, measureable parameter but rather a subjective measure on how spherical or how round a particle is. We adopted the grain shape description that is used in sedimentology for our study. We also added a paragraph on how we obtained the weights.

**Note:** when checking again the values of the weights we noticed that there was a confusion between the publications so that the weights reported in the table and in our scripts were mixed up. Now we correctly take into account the amount of friction increase for each of the publications. However, there is only a minor influence on the average quality score. The general trends remain the same. To explain, the weighting is done by comparing the increase in friction from measured or calculated

values in the respective publications. If a parameter has a stronger weighting, it has a stronger influence on the friction.

A slightly modified version of Figure A1 now is part of the discussion and referenced where correlation coefficients are mentioned. A description of the two correlation coefficients, their uncertainties and why they were chosen is included in the discussion, too.

**Grain Characteristics**

**Reviewer 1:** "Results section: Reorganizing to present the grain characteristics first would provide context for how the different materials compare to one another, which could help the reader more readily digest the differences in healing rates and reactivation. This reorganization could also improve the flow from results to discussion."

**Reviewer 2:** "I would probably move section 3.3 Grain characteristics (and Figure 6) before the other results sections to give the reader an idea of the materials of this study and differences/similarities between them. I also think this would be helpful to more easily link grain and frictional properties. Another option could be to add a brief material section where all materials tested are introduced. This might also help clarifying what has been tested in this work and what has been reanalysed (see previous comment)"

We moved the section to the beginning of the results section and included a more detailed description of the samples in the introduction and Table 1.

**Implications and Application of Results**

**Reviewer 1:** "The discussion section addresses many valid points relating to fault reactivation in basin inversion models. The findings of this manuscript may be more broadly applicable. More specifically, lines 382-389 include a key interpretation that is relevant to many analogue models. The authors introduce potential implications in the introduction that could be addressed in the discussion section to further emphasize the impact of the findings on a range of analogue models that use granular materials. Two examples of such discussion topics are 1) How could healing impact fault behavior within models with erosion/sedimentation that are stopped to remove/add material? and 2) How could consolidation and healing time impact repeatability? "

**Reviewer 2:** "The introduction section mentions the impact of healing time on the repeatability of the experiments, which is a crucial aspect for modellers working with analog materials. The introduction also mentions the impact of healing on models with erosion and sedimentation. I think these two points could be discussed more to strengthen the importance of the findings of this paper. This would be also of interest for modellers working with granular materials but not specifically on basin inversion."

We have created a new subsection in the first section of the discussion to address these issues. Additionally, we added another point to the conclusion to emphasize this result of the manuscript. The new text is:

*In the general context of analogue models, time dependent change of fault strength has further effects whose influence is small but should nevertheless be taken into account. There is a class of analogue models, also common in basin inversion, where frequent interrupts are inevitable. Typical examples are models with concurrent sedimentation or erosion where material is added or removed during a hold phase (e.g., Bonnet et al., 2007; Graveleau and Dominguez, 2008; Molnar and Buiter, 2022). With the increased use of computer tomography to screen experiments comes another type of model that must be interrupted at regular intervals. As mentioned in section 4.1.2, interruptions that are shorter than one hour should not have a great effect. The change in normal stress due to the addition or removal of overburden should have a much stronger effect than the restrengthening due to time consolidation. During the addition of material and the associated preparation of the model surface, a hardening of the fault zone could occur. For example, smoothing the surface with a spatula or ruler could exert additional stress on the fault zone and thus promote compaction. However, this effect is already commonly known for such models and has a much greater impact than the healing.*

*Repeatability of models can become an issue when the timing of the individual model runs is not the same. In recent years, more attention has been paid to the repeatability of analogue experiments. For example, Santimano et al. (2015) were able to demonstrate that various parameters such as fault*

*length and activity are intrinsically variable. Likewise, in many studies, a repetition of simple experiments is carried out to quantify the influence of preparation, setup and climatic conditions. Time strengthening brings another factor into play here. For example, models that have the same setup, but run for different lengths of time due to the duration, could have differences in fault activity or fault orientation. An experiment that is left overnight or for a longer period of time tends to have a higher strength than a model that is deformed within a short period of time. There is anecdotal evidence of analogue models that have failed to reach the expected results when the material has been sitting in the box for too long. For example, a typical sandbox experiment is performed more frequently at the GFZ Potsdam's analogue Laboratory (HelTec) for demonstration purposes. It was found that for a representative result, the time difference between preparation and execution should exceed one day. Therefore, especially in experiments with heterogeneities, for example due to faults or precut surfaces, care should be taken that the duration of inactivity of the faults is approximately the same in repeated experiments.*

**General Comments**

**Reviewer 1:** "Formatting/grammar: Some paragraphs have indentations, but others do not."

We have used the EGU Solid Earth LaTeX template and suspect that this is a formatting issue coming from their side and will be removed during the production process.

**Reviewer 1:** "Discussion section: Sections 4.1 and 4.4 are closely related and the reader could benefit from the two sections being next to each other or integrated."

We have subdivided section 4.1 into three smaller subsections containing the previous sections 4.1 and 4.4 as well as a newly created section on 'General Impact of Time Consolidation on Analogue Models". The latter addresses the comments on "Implications and Application of Results" mentioned above.

**Reviewer 1:** "What is the reason to use Pearson and Spearman correlation coefficients for the comparisons between healing rate/friction/cohesion and the qualitative properties?"

To address this point we added the following text to the discussion of the correlations: *The Pearson correlation coefficient assumes a linear correlation between two variables which might not always be present. Therefore we also use the Spearman correlation coefficient which is independent of the underlying distribution. For both parameters the p-value is determined which gives an indication of the significance of correlation. Generally, if $p < 0.05$ one assumes that the found correlation is not possible through random permutation of the original data and therefore statistically significant. It should be noted, however, that in this manuscript the quantification of the quality score is subjective, while the friction parameters including their errors are based on real measured values. Thus, the use of correlation coefficients is only an aid to identify trends more easily in the set of measurements and samples.*

**Reviewer 1:** "What is uncertainty for each quality score, and how does such uncertainty impact the correlation coefficients? In other words, how much weight should be placed in the correlation coefficients and p values?"

To address this point we added the following text to the methods section: *The quality score is a qualitative measure that only gives a general tendency of material behavior and follows an arbitrary scale. We use this scale to turn a qualitative observation into a quantity. Here, the purely subjective observation of the grains, without capturing a truly measurable quantity, represents a major source of error. The quality score is therefore only a theoretical support to find possible correlations between the parameters. In no way should the score be seen an objective quantity. It could serve as a preliminary stage for future measurable variables and give a rough direction.*

**Reviewer 1:** "Figure A1 supports the conclusion regarding the correlation between sphericity and healing rates and friction coefficients. As such, Figure A1 should be included in the main text."

We moved the figure to the discussion section.

**Figure and Table Comments**

**Figure 3, Reviewer 1:** "Including a simple note on what the numbers next to the material name mean could add clarity. Indicating which materials were included in Klinkmüller et al., 2016 is a nice touch."
    We added an explanation of the numbers to the figure caption.

**Figure A1, Reviewer 1:** "Needs a legend. It would be beneficial to include in the main text since the discussion relies on the correlations that the plots demonstrate."
    We moved the figure to the discussion section. The color scheme is the same as for all other plots, nevertheless we added a legend to the top. Additionally, we now show healing rate, compaction rate and all friction and cohesion coefficients side by side. The scaling of the y-axis is now shared between the individual parameters. A trendline has been added to indicate the behavior of the parameters.

**Inline Comments**

**L186 and 187, Reviewer 1:** "Add spaces between numbers and units"
    We checked every mention of numbers and units and added spaces where needed.

**L16, 22, 42, 56, 116, 121, 133, 221, 235, 238, 245, 288, 307, 341, Reviewer 1:** "Add commas before 'which'"
    We checked every occurrence of 'which' and added commas or rephrased the sentences to remove 'which' and shorten them.

**L6-7, Reviewer 1:** "By older faults do you mean faults that have been inactive longer? This sentence is not entirely clear."
    Yes. We have rephrased the sentence to: *Faults that have been inactive for a long time therefore have a higher strength than younger faults.*

**L29, Reviewer 1:** "Remove 'e.g.'"
    Done.

**L40-42, Reviewer 1:** "Use of 'it' is slightly confusing. I presume you mean analogue models?"
    We agree. 'it' has been changed to 'Analogue models are built'

**L51, Reviewer 1:** "I would recommend citing Reber et al. 2020, which covered a range of analogue modelling benefits, applications, and material properties `https://doi.org/10.1016/j.earscirev.2020.103107`"
    We agree and also added some more references to other publications.

**L112, Reviewer 1:** "Change 'materials' to 'material'"
    Done.

**L116, Reviewer 1:** "Remove 'and' following 'fault zone'"
    We have split and this sentence in two.

**L138, Reviewer 1:** "vl doesn't seem to appear elsewhere. Maybe it doesn't need a variable assigned."
    We now mention the loading velocity $v_L$ and keep the variable for comparison with the data publication where this is mentioned more often.

**L208, Reviewer 1:** "Remove repeated 'increase of hold time'"
    Done.

**L241-243, Reviewer 1:** "Complex sentence that could be clarified"
    We have split this sentence in two and made minor adjustments to make it clearer.

**L267, Reviewer 1:** "Change 'which' to 'and' and remove 'therefore'"
    We have rephrased this sentence slightly to improve readability.

**L296, Reviewer 1:** ”Change 'which' to 'that'”
    Done.

**L306, Reviewer 1:** ”Insert comma after gray”
    Done.

**L359, Reviewer 1:** ”Out of curiosity, would you expect that healing may have a larger role for materials that exhibit stick-slip behavior (e.g., Utrecht sand?)?”
    This is a very good question. In short: Yes.
There are several factors that have to be taken into account for a material to stick-slip. First and foremost, the material itself is not enough to define if stick-slip occurs. For that the whole system (machine + sample or fault zone + lithosphere) has to be taken into account. A higher healing rate $b$ has two effects (Dieterich, 2007):

- The higher the healing rate $b$ the greater the difference $a - b$ with the direct effect $a$ being constant. In theory, this makes the material more velocity weakening and thus more unstable.

- With higher $b$ the critical stiffness increases. Therefore, a system that was previously stable, such as the ring shear tester, now needs to have a higher stiffness to sustain stable sliding. As a result the system shows a higher tendency for stick-slips.

There are however conditions where $b$ can be smaller than $a$ and stick-slip still occurs (Dieterich, 2007). We have added another paragraph concerning this to the discussion.

**References**

Bonnet, C., Malavieille, J., and Mosar, J. (2007). Interactions between tectonics, erosion, and sedimentation during the recent evolution of the alpine orogen: Analogue modeling insights. *Tectonics*, 26(6):n/a–n/a.

Dieterich, J. (2007). Applications of rate-and state-dependent friction to models of fault slip and earthquake occurrence. *Treatise on Geophysics*, 4:107–129.

Graveleau, F. and Dominguez, S. (2008). Analogue modelling of the interaction between tectonics, erosion and sedimentation in foreland thrust belts. *Comptes Rendus Geoscience*, 340(5):324–333.

Klinkmüller, M., Schreurs, G., Rosenau, M., and Kemnitz, H. (2016). Properties of granular analogue model materials: A community wide survey. *Tectonophysics*, 684:23–38.

Molnar, N. and Buiter, S. (2022). Analogue modelling of the inversion of multiple extensional basins in foreland fold-and-thrust belts.

Rudolf, M., Rosenau, M., and Oncken, O. (2021). Ring shear and slide-hold-slide test measurements for soda-lime glassbeads of 300-400µm diameter used at the helmholtz laboratory for tectonic modelling, potsdam, germany.

Santimano, T., Rosenau, M., and Oncken, O. (2015). Intrinsic versus extrinsic variability of analogue sand-box experiments – insights from statistical analysis of repeated accretionary sand wedge experiments. *Journal of Structural Geology*, 75:80–100.